# Image Entropy and Numeric Representation for MRI Semantic Segmentation

## Abstract

Deep learning has made major strides in medical imaging segmentation in the last several years for its automated feature extraction. This model fitting process is susceptible to over-fitting, and can benefit from sparsity. Here, we show theoretical and experimental potential of using low-entropy images as sparse input to improve deep learning driven tissue segmentation, using tumor and heart segmentation problems as exemplary cases.

**Keywords:** Segmentation, Numerical Representation, Image Entropy, Deep-Learning, U-Net, MRI

## 1. Introduction

Deep neural networks have taken center stage for their ability to take highly complex data as input, learn their own feature representation, and successfully converge to a solution. Because of this, manual feature engineering techniques are largely ignored in contexts where deep neural networks have proven successful. However, the convolutional layers often doing the automated feature engineering remain high variance models [(Menart, 2020)]. Therefore it stands to reason that in situations with few samples relative to a very large number of modelling parameters (as is often the case in medical imaging) that more effective training could be achieved with some manual feature engineering. Here, we propose reducing the numeric range of the input as a way to have feature/signal "sparsity" without hampering the network's automated feature engineering. We argue that sparsity is akin to entropy, and that we can reduce image entropy by constraining the numerical input range of the images. We study the effect of reducing input range with a couple of standard medical imaging segmentation problems, firstly tumor segmentation, and afterwards, for comparison, segmentation of the left atrium of the heart. We do not present these methods as a catch-all for reducing the image entropy or improving all medical imaging/segmentation problems, but as a proof-of-principal that when working with deep learning architectures and small datasets, controlling the entropy of the input image will have an effect on model performance and should be considered.

## 2. Theory

We define sparsity as the ability to concentrate the energy function of a signal or model in as few coefficients as possible. Energy refers to a function in a Gibbs measure, E(x), which moves the space of states to real numbers. As the space of states, or energy, is concentrated in fewer coefficients the signal is considered to be more compressible and therefore inherently

sparser. This is made clear with the following two examples adapted from (Pastor et al., 2015). Consider the random variable $X\epsilon\{x^1, x^2\}$ with probability distribution $p = (p_1, p_2)$. First, assume $p_1 > p_2$, with $p_1, p_2 >= 0$ and $||p|| = 1$. Then if $p_1$ increases, it is obvious then $x^1$ is more likely to appear, so the compressiblity, or sparsity, of $p$ increases, and the uncertainty, or entropy, of $X$ must decrease. Second, assume, $p = (p_1, p_2) = (1, 0)$, so the distribution represents a constant random variable. If $p_2$ increases at all, then $x^1$ is no longer the unique possible outcome and the compressibility of $p$ must decrease with the increase in uncertainty in $x$. We can now conclude that reducing the image entropy results in a sparse representation of the most predicable intensity values. Effectively a smaller input space will help control model variability by placing a strong inductive bias over the data, namely that a solution must be found in a low information environment, and to achieve this the input data space should be low information with few possible states.

## 3. Material and methods

### 3.1. Data

Brain tumor tissue segmentation presents a particularly challenging problem, and a useful testing ground for experimental methodologies. We used the 2021 BraTS publicly available training dataset. All details regarding the dataset can be found in the latest BraTS summarizing paper [(Bakas et al., 2018)]. Each data point was saved as one of three input types: a reference 16-bit image, then a simple numeric reduction normalizing all values between 0-255 for an 8-bit image, and an extreme reduction to 3-bits of information, by z-scoring the image and truncating to the nearest integer. For the second example, we used the heart segmentation task from the Medical Imaging Decathlon. The target region of interest for the task is the left atrium of the heart. The data was originally acquired as part of the 2013 Left Atrial Segmentation Challenge [(Tobon-Gomez et al., 2015)].

### 3.2. Experimental Procedure

As high variance models, deep neural networks are sensitive to the sampling variability of the training set. To address this, we experiment with only simple 3D U-Net models [(Çiçek et al., 2016)] since the model is small it allows efficient permutation testing of experimental conditions. And while simple, the 3D U-Net is still the backbone of most state-of-the-art medical segmentation techniques [(Siddique et al., 2021)], and should still give insight on the effect of entropy on the input. We bootstrapped the dataset to give 10 permutations for the tumor segmentation, and 19 for the heart segmentation. For both, the training objective function was a simple Dice co-efficient loss between the output and the target. After building a confidence interval around the model estimates, we apply a one-way ANOVA for each tissue type combined with non-parametric bootstrapping over the dataset to test the mean differences between our experimental conditions while accounting for sampling effects in the training set. The models were evaluated using the Dice co-efficient scores between the predicted segmentation and the ground truth labels.

## 4. Results

All tumor tissue ANOVAs showed significant differences between input data representations, at $p < 0.001$. On average the lower the input information the better the model preformed. This is visualized in figure 1. The ANOVA to asses the left atrium of the heart segmentation resulted in a modestly significant difference between input types, with $p = 0.037$, again with the most reduced input preforming the best, and displayed in figure 2.

## 5. Conclusion

We shows that reducing image entropy may help with complex segmentation tasks (tumor), but is of less use when the task is already simple (heart). Despite the aforementioned limitations, this is an exciting result that deserves further investigation as it could be *low hanging fruit* to improve data-hungry segmentation methods in medical imaging.

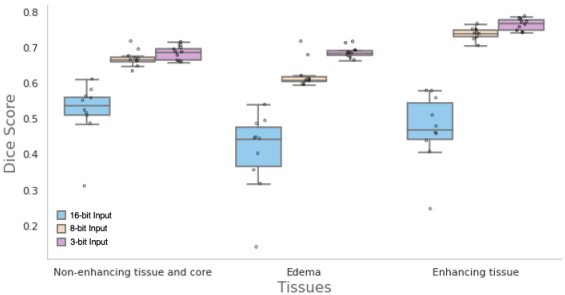

Figure 1: Bootstrap generated tissue map segmentation estimates on the independent test data. Lower input information greatly increased the corresponding model's Dice coefficient estimates.

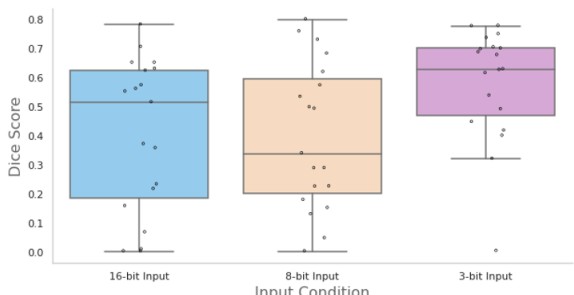

Figure 2: Bootstrap generated left atrium segmentation estimates on the independent test data. Lowering the input information mildly increased Dice coefficient estimates.

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
