# OpenReview forum: "Image Entropy and Numeric Representation for MRI Semantic Segmentation"
_MIDL.io/2023/Short_Paper_Track — MIDL 2023 Short paper track Poster_

### Official Review · Reviewer_3J6v · 2023-04-10
**Test MRI segmentation performance with image entropy**

**Rating:** 4
**Confidence:** 5

**Review:**

This paper presents the experimental results of using low-entropy images (reduce bits) as sparse input for deep learning-driven tissue segmentation
The advantages of the paper include:
+ This paper applies the different configurations of bits on BRATs dataset.
+ Box plots are present to show the variations
The limitation of the paper includes:
- Many papers have already discussed the impacts of bits on medical image segmentation
- The situation is case by case. Brain tumor is a relatively simple task. The conclusion might not hold for detailed brain tissue segmentation on MRI
- The technical innovation is limited as a 3D U-Net is applied to a different dataset.
- Qualitative results might be helpful for presenting the results.

---

### Official Review · Reviewer_CCW9 · 2023-04-10
**Unexpected but weird but intriguing results**

**Rating:** 7
**Confidence:** 3

**Review:**

The authors show how segmentation accuracy with a standard 3D U-net can greatly improve when the input images are quantized to a small number of bits. This improvement is *massive* (20 Dice points) in one of the tasks. The authors' hypothesis is that lower entropy images facilitate learning in low-data regimes. However, It is difficult not to wonder whether there is a problem with the data normalization or some other component of the setup. Either way, I think that this paper could spark interesting discussion at the conference.

PS: a (very minor) slap on the wrist to the authors, for withholding their names & institutions on the pdf, which saves space (maybe they were not aware that short papers were single-blind - in which case, the slap on the wrist is for not reading the instructions!)